Molecular Biology and Physiology

# GapMind: Automated Annotation of Amino Acid Biosynthesis

Morgan N. Price,[a] Adam M. Deutschbauer,[a,b] Adam P. Arkin[a,c]

[a]Environmental Genomics and Systems Biology, Lawrence Berkeley National Laboratory, Berkeley, California, USA
[b]Plant & Microbial Biology, University of California, Berkeley, California, USA
[c]Department of Bioengineering, University of California, Berkeley, California, USA

**ABSTRACT** GapMind is a Web-based tool for annotating amino acid biosynthesis in bacteria and archaea (http://papers.genomics.lbl.gov/gaps). GapMind incorporates many variant pathways and 130 different reactions, and it analyzes a genome in just 15 s. To avoid error-prone transitive annotations, GapMind relies primarily on a database of experimentally characterized proteins. GapMind correctly handles fusion proteins and split proteins, which often cause errors for best-hit approaches. To improve GapMind's coverage, we examined genetic data from 35 bacteria that grow in defined media without amino acids, and we filled many gaps in amino acid biosynthesis pathways. For example, we identified additional genes for arginine synthesis with succinylated intermediates in *Bacteroides thetaiotaomicron*, and we propose that *Dyella japonica* synthesizes tyrosine from phenylalanine. Nevertheless, for many bacteria and archaea that grow in minimal media, genes for some steps still cannot be identified. To help interpret potential gaps, GapMind checks if they match known gaps in related microbes that can grow in minimal media. GapMind should aid the identification of microbial growth requirements.

**IMPORTANCE** Many microbes can make all of the amino acids (the building blocks of proteins). In principle, we should be able to predict which amino acids a microbe can make, and which it requires as nutrients, by checking its genome sequence for all of the necessary genes. However, in practice, it is difficult to check for all of the alternative pathways. Furthermore, new pathways and enzymes are still being discovered. We built an automated tool, GapMind, to annotate amino acid biosynthesis in bacterial and archaeal genomes. We used GapMind to list gaps: cases where a microbe makes an amino acid but a complete pathway cannot be identified in its genome. We used these gaps, together with data from mutants, to identify new pathways and enzymes. However, for most bacteria and archaea, we still do not know how they can make all of the amino acids.

**KEYWORDS** amino acid biosynthesis, gene annotation, high-throughput genetics

Genome sequences are available for tens of thousands of microbes. For most of these microbes, little is known about their physiology other than the condition under which they were isolated. If the microbe was isolated using a complex substrate, such as yeast extract, then nothing is known about its nutritional requirements. To understand the ecological roles or the potential uses of these microbes, it is important to understand their growth requirements, which, in principle, could be predicted from their genome sequences. Specifically, we will focus on whether a microbe can synthesize the 20 standard amino acids.

Although some comparative genomics tools try to predict which amino acids a microbe can synthesize (1, 2), the predictions are not at all reliable (3). For instance, when we tested the Integrated Microbial Genomes tool (1) with bacteria that can grow in minimal media, we found that, on average, these bacteria were predicted to be

Address correspondence to Morgan N. Price, morgannprice@yahoo.com, or Adam P. Arkin, aparkin@lbl.gov.

GapMind: automated annotation of amino acid biosynthesis http://papers.genomics.lbl.gov/cgi-bin/gapView.cgi?orgs=orgsFit&set=aa&orgId=FitnessBrowser__Btheta

auxotrophic for six amino acids, even though they can make all of them (3). Alternatively, auxotrophies can be identified using genome-scale metabolic models (4), but accurate models are not available for most taxa. As far as we know, accurate and automated prediction of auxotrophies has only been successful for well-studied taxa, such as *Enterobacteria* or *Pseudomonas* (4). In a study of 40 diverse gut bacteria that grow in defined media, automatically generated metabolic models failed to predict growth for 30 of them (5).

Predicting growth requirements automatically is challenging for several reasons. First, many bacteria do not use the standard biosynthetic pathways from *Escherichia coli* or *Bacillus subtilis* that are described in textbooks. These variant pathways are often missing from the databases that automated tools rely on (3, 6). Variant pathways and variant enzymes continue to be discovered, so accurate prediction of microbial growth capabilities from genome sequences alone may not yet be possible (3).

Second, predicting enzymatic activity from a protein's sequence is challenging if the sequence is very different from that of any protein that has been studied experimentally. To increase their coverage, comparative tools often rely on databases of annotated proteins, including annotations for proteins that have not been studied experimentally. Unfortunately, many of the enzyme annotations in databases such as GenBank, KEGG, or SEED are incorrect (7, 8). Another problem is that comparative tools often rely on identifying best hits, which does not work well for fusion proteins or split proteins. For instance, if a protein is a fusion of *X* and *Y* and its best hit is *X*, then it might be annotated as *X* and *Y* might appear to be absent.

We built a tool, GapMind, to reconstruct and annotate amino acid biosynthesis pathways in prokaryotic genomes. Given our limited understanding of biosynthetic pathways and the challenges of automated annotation, GapMind does not predict whether a biosynthetic capability is present or not. Instead, it identifies the most plausible pathway for making each amino acid based on current knowledge, and it highlights potential gaps. For instance, if a diverged candidate for a step is identified, then it is labeled as medium confidence; if this step is part of the most likely pathway, then it will be highlighted. The user can examine the results and decide if the pathway is likely to be present or not.

To try to ensure that GapMind's results can be traced to experimental data on the function of similar proteins, GapMind relies primarily on similarity to experimentally characterized proteins. GapMind does not use best hits, and it handles fusion proteins and split proteins correctly. GapMind has a Web-based interface (http://papers.genomics.lbl.gov/gaps) and takes about 15 s per genome to run.

GapMind's database includes dozens of variant biosynthetic pathways and enzymes. To identify additional variants, we tested GapMind on 35 bacteria that grow in defined media and for which large-scale genetic data are available. Based on the genetic data, we incorporated two variant pathways and dozens of diverged enzymes into GapMind's database.

Nevertheless, many variant pathways and enzymes remain to be discovered. Thus, GapMind also includes a database of "known gaps": steps that appear to be missing, yet the organism does grow in minimal media. If a genome of interest appears to lack a step that is a known gap in a similar organism (that can grow in minimal media), then GapMind marks the step as a known gap. This way, the user can see that the gap may be due to an as-yet unknown enzyme or pathway.

(This article was submitted to an online preprint archive [9].)

## RESULTS

**How GapMind works. (i) The amino acid biosynthesis pathways included in GapMind.** GapMind describes the biosynthesis of 17 amino acids and of chorismate, which is a precursor of the aromatic amino acids. GapMind does not include the biosynthesis of the other three amino acids (alanine, aspartate, or glutamate), because each of these is formed by the transamination of an intermediate from central metabolism (pyruvate, oxaloacetate, or $\alpha$-ketoglutarate, respectively). Amino acid transami-

**A. Example pathway**

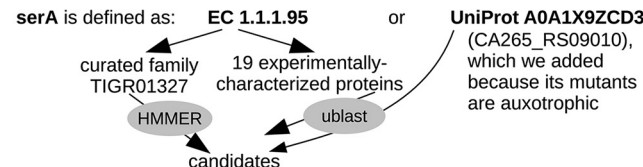

serine

**B. Example step**

serA is defined as: **EC 1.1.1.95** or **UniProt A0A1X9ZCD3** (CA265_RS09010), which we added because its mutants are auxotrophic

curated family TIGR01327    19 experimentally-characterized proteins

HMMER    ublast

candidates

**C. Confidence levels for candidates from ublast**

**High confidence:**
- Aligns at 40% identity
- Covers 80% of the characterized protein
- And candidate is less similar to proteins with other functions

candidate

characterized protein

Medium confidence:
- Aligns at 30% identity and 80% coverage and less similar to proteins with other functions
- Or, 40% identity and 70% coverage (even if similar to proteins with other functions)
- Or, similar to a curated (uncharacterized) protein from Swiss-Prot

**Low confidence:**
- Aligns at 30% identity and 50% coverage (even if similar to proteins with other functions)

**D. Confidence levels for candidates from HMMER**

**High confidence:**
- Scores above the trusted cutoff
- Alignment covers 80% of the HMM
- And candidate is <40% identical to proteins with other functions

Medium confidence:
- Scores above the trusted cutoff

**FIG 1** How GapMind works. (A) A pathway with no variants. (B) The definition of a step. (C) Confidence levels for candidates from ublast. (D) Confidence levels for candidates from HMMER.

nases are often nonspecific and annotating their precise substrates is challenging (3), so we assume that enzymes that catalyze these three transamination reactions are present and that the amino acid can be produced.

Most of the pathways in GapMind were taken from the MetaCyc database of metabolic pathways and enzymes (10). In addition, a few variant pathways that are not currently in MetaCyc are included in GapMind. These additional pathways are listed in Text S1 in the supplemental material or are described below.

We tried to include all known pathways for amino acid biosynthesis that begin with intermediates in central metabolism and that occur in bacteria or archaea. Because most free-living bacteria and archaea can probably make all 20 standard amino acids (3), we also allow pathways to use other amino acids as starting points. For example, many microorganisms synthesize cysteine from serine and sulfide.

Our primary goal is to understand how a microbe might be able to grow with minimal nutrients, so we did not include pathways that correspond to unusual nutritional requirements. For example, GapMind does not include glycine synthesis from glycolate (11) or cysteine biosynthesis from sulfocysteine. GapMind also does not include cysteine biosynthesis from serine and methionine, because prototrophic organisms would use the simpler reverse transsulfuration pathway from serine and homocysteine. A few pathways with uncertain occurrence in bacteria or archaea were also omitted (Text S2). On the other hand, we included isoleucine biosynthesis from propionate, because propionate is an end product of fermentation and need not be a nutritional requirement (12).

**(ii) How GapMind represents pathways.** In GapMind, each pathway is broken down into a list of steps (Fig. 1A). For heteromeric enzymes, each subunit is treated as a separate step. Alternate pathways are indicated by alternate lists of steps. To simplify

the analysis of pathways with many variants, a pathway can include subpathways as well as steps. GapMind currently has 45 subpathways.

Most steps are described using enzyme commission (EC) numbers or terms (Fig. 1B). To list the proteins that are known to carry out each step, GapMind compares EC numbers or terms to the curated descriptions of over 100,000 experimentally characterized protein sequences. The biggest source of characterized proteins is Swiss-Prot (13). GapMind also describes some steps using protein families from TIGRFam (14) or Pfam (15) or by using proteins that we curated based on published papers or genetic data (see below).

Altogether, GapMind represents amino acid biosynthesis with 149 steps. These steps are associated with 1,821 different characterized proteins (including 99 proteins that we curated), 140 TIGRFams, and 4 Pfams.

**(iii) How GapMind identifies candidates.** Given the proteins and families that are associated with each step, GapMind searches a genome of interest for candidates (Fig. 1B). To search for similar proteins, it uses ublast (16); to search for members of families, it uses HMMER (17). GapMind then uses ublast to check if these candidates are similar to characterized proteins that have other functions. At this stage, GapMind compares the candidates to all characterized proteins, not just those involved in amino acid biosynthesis.

**(iv) Confidence levels for candidates, steps, and pathways.** Intuitively, a protein is a high-confidence candidate for a step if it is sufficiently similar to a protein that is known to carry out that step (Fig. 1C and D). For high-confidence candidates, GapMind requires 40% identity to a characterized protein with 80% coverage or a match to a curated family with 80% coverage. We chose 40% identity as a threshold, because more distantly related enzymes often have different substrates (18). The 80% coverage requirement should ensure that all of the domains required for catalysis are present. GapMind also requires that the candidate be more similar to the characterized protein than to any protein known to have another function; this should ensure that most of the high-confidence candidates act on the correct substrates.

To identify moderate-confidence candidates, GapMind uses lower thresholds: down to 30% identity with 80% coverage (if not more similar to a protein with another function), 40% identity with 70% coverage (regardless of similarity to other proteins), or a hit to an HMM above the trusted cutoff (regardless of coverage or similarity to other proteins). To identify moderate-confidence candidates, GapMind also uses similarity to experimentally uncharacterized proteins from bacteria and archaea that have curated enzyme annotations in Swiss-Prot. This adds another 45,090 sequences to the database that GapMind considers.

Candidates with at least 30% identity to a characterized or curated sequence and at least 50% coverage are considered low confidence. These low-confidence candidates are shown on the GapMind website because they may be useful for filling gaps in amino acid biosynthesis pathways.

Given the confidence levels for the candidates, GapMind computes confidence levels for steps and pathways and finds the highest-confidence pathway for synthesizing each amino acid. The confidence of a step is the highest confidence of any candidate for that step. Steps are considered low confidence even if they have no candidates at all. This ensures that pathways are considered even if they have gaps due to as-yet-unknown variant enzymes. The confidence of a pathway is the lowest confidence of any step in that pathway.

**(v) Fusion proteins.** GapMind's approach automatically handles fusion proteins. GapMind scores steps independently of each other, so a protein can be a high-confidence candidate for more than one step. In addition, when GapMind tests if a candidate is similar to a protein with another function, it ignores hits that are outside the relevant region of the candidate. Thus, if two enzymes are fused into one protein, GapMind will usually link the protein to both steps. In contrast, if genes were annotated using best hits, the fusion protein would have just one best hit and would be a candidate for, at most, one step. For example, as shown in Fig. 2A, HSERO_RS20920

**A. Fused AroL-AroB**

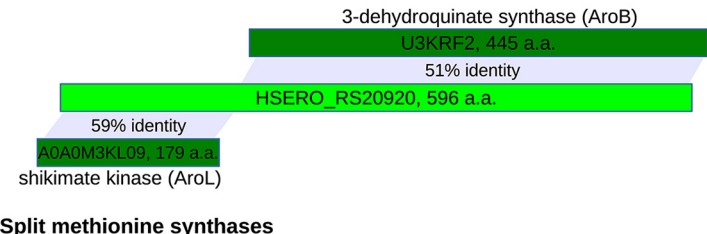

**B. Split methionine synthases**

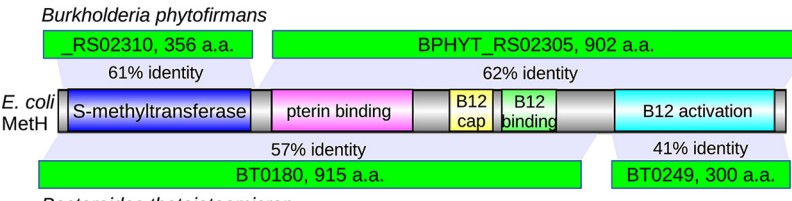

**FIG 2** GapMind handles fusion proteins and split proteins. (A) HSERO_RS20920 from *Herbaspirillum seropedicae* SmR1 is a fusion of AroL and AroB (shown with Swiss-Prot identifiers). (B) Split candidates for vitamin $B_{12}$-dependent methionine synthase (MetH) in *Burkholderia phytofirmans* PsJN and *Bacteroides thetaiotaomicron* VPI-5482. a.a., amino acids.

from *Herbaspirillum seropedicae* SmR1 is a fusion of shikimate kinase (AroL) and 3-dehydroquinate synthase (AroB). The similarity of the C-terminal region to 3-dehydroquinate synthase scores more highly than the similarity of the N-terminal region to shikimate kinase, so a best-hit approach might to annotate the entire protein as AroB. In contrast, when testing whether HSERO_RS20920 is a high-confidence candidate for AroB, GapMind ignores the alignments of the C-terminal part of the protein to 3-dehydroquinate synthases. Thus, HSERO_RS20920 is a high-confidence candidate for both AroB and AroL.

**(vi) Split proteins.** GapMind also looks for "split proteins," where a multidomain protein is split into two pieces. For instance, in *Escherichia coli*, the vitamin $B_{12}$-dependent methionine synthase MetH is a single protein with five domains. In *Burkholderia phytofirmans* PsJN, BPHYT_RS02305 contains the pterin-binding domain, the two domains involved in binding vitamin B12, and the domain for the reactivation of vitamin B12, while BPHYT_RS02310 contains the *S*-methyltransferase domain (Fig. 2B). In *Bacteroides thetaiotaomicron* VPI-5482, methionine synthase is split in a different way, with the reactivation domain in one protein (BT0249) and the other four domains in another protein (BT0180) (Fig. 2B). GapMind automatically joins these proteins together based on the nonoverlapping alignments of two pieces to the same characterized protein (Fig. 2B). However, GapMind cannot detect more complicated arrangements, such as the splitting of methionine synthase from *Phaeobacter inhibens* into three proteins, together with a nonhomologous system for the reactivation of vitamin $B_{12}$ (3). These proteins from *P. inhibens* are also rather diverged from other methionine synthases, so we added a subpathway to describe the three-part methionine synthase.

**Expanding GapMind's database using genetic data.** To improve GapMind, we tested it on 35 diverse bacteria that can make all 20 amino acids and for which we have large-scale genetic data from pools of transposon mutants (8, 19–21). These bacteria are listed in Data Set S1. We previously used genetic data for 10 of these bacteria to fill some gaps in their amino acid biosynthetic pathways (3), and these previously filled gaps were already incorporated into GapMind's database of characterized proteins. Nevertheless, across all the pathways, the average bacterium had 3.7 gaps, or steps that were on the best path but were not high confidence. These gaps included 0.8 low-confidence steps and 2.9 medium-confidence steps per bacterium.

We used genetic data from these 35 bacteria growing in minimal media to identify the genes for many of the missing steps. We found evidence for two poorly studied

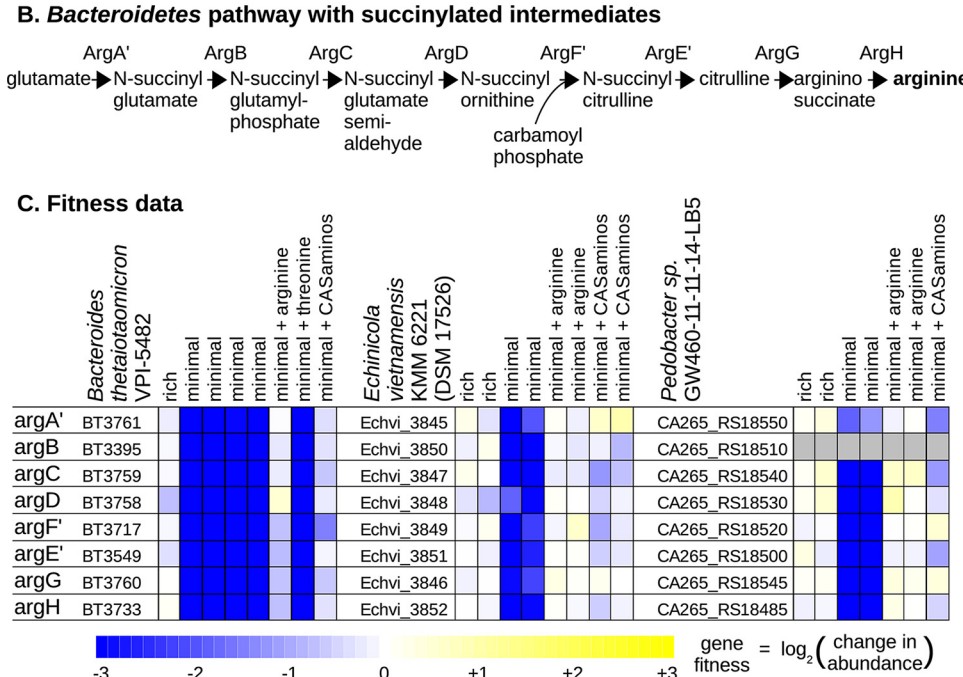

**FIG 3** Arginine biosynthesis with succinylated intermediates. (A) The standard pathway. Protein names are from *Escherichia coli* or *Bacillus subtilis*. The formation of carbamoyl phosphate (catalyzed by CarAB) is not shown. (B) The pathway in *Bacteroides* and in other *Bacteroidetes*. (C) Fitness data from *Bacteroides thetaiotaomicron* VPI-5482, *Echinicola vietnamensis* KMM 6221 (DSSM 17526), and *Pedobacter* sp. strain GW460-11-11-14-LB5 (from references 8 and 20). Each fitness value is the $\log_2$ change in the abundance of the mutants in a gene during an experiment. Each experiment went from an optical density at 600 nm of 0.02 to saturation (usually 4 to 8 doublings). Fitness values for CA265_RS18510 were not estimated, because mutants of this gene were at low abundance in the starting samples.

pathways and confirmed dozens of candidates that were divergent from previously characterized proteins.

**(i) Arginine synthesis with succinylated intermediates.** Our preliminary version of GapMind identified four gaps in arginine synthesis in *Bacteroides thetaiotaomicron* VPI-5482. First, at the beginning of the pathway, no candidates for *N*-acetylglutamate synthase (ArgA or ArgJ) were identified.

Second, neither ornithine carbamoyltransferase (ArgI) nor acetylornithine carbamoyltransferase was identified with high confidence. BT3717 was identified as a candidate for acetylornithine carbamoyltransferase, but BT3717 is nearly identical to a characterized enzyme from *Bacteroides fragilis* that acts on *N*-succinylornithine instead (22). In fact, *B. fragilis* was proposed to synthesize arginine via succinylated intermediates (22) instead of acetylated intermediates (Fig. 3A and B). Furthermore, ArgB from *B. fragilis* is an *N*-succinylglutamate kinase, not an *N*-acetylglutamate kinase (23), which confirms that *B. fragilis* uses succinylated intermediates. Unfortunately, as of June 2019, arginine synthesis with succinylated intermediates was not described in any of the standard databases (Swiss-Prot, MetaCyc, KEGG, or SEED), and BT3717 was misannotated in Swiss-Prot as acetylornithine carbamoyltransferase instead of succinylornithine carbamoyltransferase.

Third, GapMind identified BT3758 as a potential aminotransferase for converting *N*-acylglutamate semialdehyde to *N*-acylornithine but with moderate confidence, be-

cause it was less than 40% identical to any characterized enzyme. BT3758 also is 38% identical to an aminotransferase that is involved in lysine biosynthesis ([LysW]-aminoadipate semialdehyde transaminase from *Thermus thermophilus*; Swiss-Prot entry Q93R93), which creates some uncertainty about its role.

The final gap was argininosuccinate synthase (ArgH). BT3760 was identified as a moderate-confidence candidate because it is less than 40% identical to any characterized enzyme.

Using the genetic data for *B. thetaiotaomicron*, we had previously identified that BT3761 participates in arginine biosynthesis (20). BT3761 is over 50% identical to the recently discovered *N*-acetylglutamate synthase Cabys_1732 (24), but given the activities of the other enzymes from *B. fragilis*, BT3761 is probably *N*-succinylglutamate synthase. The pathway with succinylated intermediates also should have a desuccinylating enzyme, most likely an *N*-succinylcitrulline desuccinylase (Fig. 3B). The genetic data identified BT3549 as a candidate for this step (Fig. 3C); BT3549 is distantly related (under 30% identity) to succinyl-diaminopimelate desuccinylase from *Mycobacterium tuberculosis* (25), which is a similar chemical reaction. The genetic data also confirmed that the best candidates for the other steps are indeed involved in arginine biosynthesis. Specifically, these genes are important for growth in a defined minimal medium that lacks arginine but are not important for growth in rich medium or in minimal medium that was supplemented with arginine or with Casamino Acids, which includes arginine (Fig. 3C).

We also identified arginine synthesis with succinylated intermediates in two other bacteria from the phylum *Bacteroidetes* that we studied: *Echinicola vietnamensis* KMM 6221 and *Pedobacter* sp. strain GW460-11-11-14-LB5. Most of the candidate genes in these two bacteria are also important for growth in defined medium unless arginine or Casamino Acids are added (Fig. 3C).

It appears that most members of the phylum *Bacteroidetes* synthesize arginine via succinylated intermediates. When we analyzed 106 genomes from this phylum (from MicrobesOnline [26]) using GapMind, we found that 70 (66%) had a high-confidence pathway for arginine biosynthesis with no gaps. In 69 of these 70 cases, the predicted pathway was the *Bacteroides*-type pathway. The exception was *Bacteroides pectinophilus* ATCC 43243, which has been reclassified to another phylum (*Firmicutes*) in the genome taxonomy database (GTDB [27]). The amino acid sequences of the carbamoyltransferases in the 69 *Bacteroidetes* also confirm that they use succinylated intermediates. The specificity of the enzyme for *N*-succinylcitrulline or *N*-acetylcitrulline can be switched by mutating a single amino acid corresponding to position 90 of BT3717 (28). Sixty-one of these 69 *Bacteroidetes* (89%) have amino acids at that position that cause a preference for *N*-succinylcitrulline (S, P, A, or V [28]), and many of these genomes were previously predicted to encode *N*-succinylcitrulline carbamoyltransferase (28).

**(ii) Tyrosine synthesis from phenylalanine via phenylalanine hydroxylase.** Most bacteria synthesize tyrosine from chorismate via prephenate dehydrogenase or arogenate dehydrogenase, but in *Dyella japonica* UNC79MFTsu3.2, GapMind did not identify any medium- or high-confidence candidates for either enzyme. N515DRAFT_1431 was identified as a low-confidence candidate, but it appears to be a fusion of two enzymes for phenylalanine biosynthesis: chorismate mutase and prephenate dehydratase.

When we searched the genetic data from *D. japonica* for auxotrophic genes, we identified a putative phenylalanine 4-hydroxylase (PAH; N515DRAFT_3052) that is important for growth in defined media but not in rich media (Fig. 4A). This suggested that PAH is the primary route for the biosynthesis of tyrosine in *D. japonica*, but bacterial PAH are usually described as the first step in the catabolism of phenylalanine. Indeed, PAH from *D. japonica* is over 60% identical to a protein that is important for the utilization of phenylalanine as a carbon or nitrogen source (RR42_RS20365 from *Cupriavidus basilensis* 4G11; data are from reference 8). On the other hand, a biosynthetic role for bacterial PAH has been proposed before: in *Legionella pneumophila* 130b, PAH is involved in pyomelanin biosynthesis, and a PAH mutant has reduced growth in

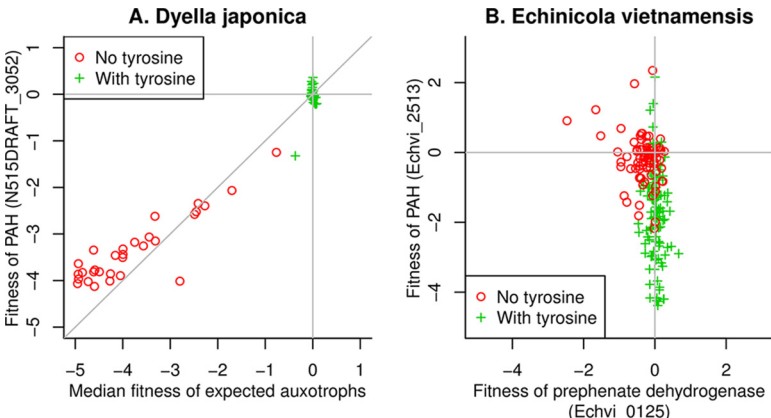

**FIG 4** Tyrosine synthesis via phenylalanine hydroxylase in *Dyella japonica* and *Echinicola vietnamensis*. (A) Gene fitness in *Dyella japonica* UNC79MFTsu3.2. The *x* axis shows the median fitness across 59 genes that are predicted to be involved in amino acid biosynthesis (by TIGRFam role [14]), and the *y* axis shows the fitness of the predicted phenylalanine hydroxylase (PAH). (B) Gene fitness in *Echinicola vietnamensis* KMM 6221 (DSM 17526) for prephenate dehydrogenase (*x* axis) and for PAH (*y* axis). In both panels, we color code experiments by whether or not tyrosine was present in the media. The experiments with tyrosine usually included it via yeast extract or Casamino Acids, while the experiments without tyrosine are in defined media with just one or no amino acids added. Lines show *x* = 0 and *y* = 0, corresponding to no effect of mutating the genes. In panel A, lines show *x* = *y*.

a tyrosine-free defined medium (29). GapMind's analysis suggests that *L. pneumophila* also lacks arogenate dehydrogenase and prephenate dehydrogenase.

We also noticed that in *Echinicola vietnamensis*, PAH (Echvi_2513) is important for growth in some defined media (Fig. 4B). This bacterium also has a prephenate or arogenate dehydrogenase (Echvi_0125), which is important for growth in some defined media but not others (Fig. 4B). It is difficult to understand why PAH is important for fitness in defined media unless it is involved in tyrosine biosynthesis. Conversely, a biosynthetic role for PAH would explain why the prephenate or arogenate dehydrogenase appears to be dispensable in some defined media. The two genes also seem to be important for fitness under different conditions. Both genes are important for fitness under some conditions (Fig. 4B), but there are no experiments where both genes had fitness values under −1 (which corresponds to a 2-fold reduction in the abundance of mutant strains). This suggests that the two pathways are genetically redundant. Thus, we propose that *E. vietnamensis* uses both routes for the biosynthesis of tyrosine.

Because PAH appears to be a major route for tyrosine biosynthesis in *Dyella japonica*, *Legionella pneumophila*, and *Echinicola vietnamensis*, we included PAH in GapMind. Besides phenylalanine, the other substrates for this enzyme are molecular oxygen and a pterin cofactor, so this pathway cannot function under anaerobic conditions.

**(iii) Confirming the roles of diverged enzymes.** Besides adding the two pathways described above, we used the genetic data to confirm that 43 divergent candidates that were predicted to be involved in amino acid biosynthesis were important for growth in minimal media (Data Set S2). Most of these candidates were originally considered to be moderate confidence (23/43) or low confidence (5/43), one diverged candidate was not identified by the preliminary version of GapMind, and the remaining 14 candidates had already been classified as high confidence. Most of the diverged candidates were already annotated in UniProt with the functions that we confirmed; the six exceptions are explained in Data Set S2.

In four cases, we are confident that the protein is involved in the pathway, but we cannot predict its precise activity. Tyrosine biosynthesis can proceed either from prephenate to 4-hydroxyphenylpyruvate to tyrosine (a dehydrogenase reaction followed by an aminotransferase reaction) or from prephenate to arogenate to tyrosine (an aminotransferase reaction followed by a dehydrogenase reaction). Four proteins

were similar to both prephenate dehydrogenases and arogenate dehydrogenases, and the genetic data confirmed that they were important for fitness in minimal media unless a mixture of amino acids is added. This confirms these proteins are involved in amino acid biosynthesis, but we still do not know whether they act on prephenate, arogenate, or both. In the updated GapMind, these proteins (and their homologs) are considered good candidates for either activity.

Some of the diverged candidates were essential for viability in the rich media used to construct the mutant libraries. Essentiality is common for amino acid biosynthesis genes (3) but does not give an indication as to the gene's specific role. If similar candidates from related bacteria both were essential and no other good candidates were detected, then we reasoned that the genes were probably annotated correctly and we added them to GapMind's database. Using this approach, we added another 17 proteins to GapMind's database. Nine of these proteins are candidates for steps that are essential in most bacteria: AspS2, GltX, or DapB. (AspS2 and GltX are involved in both amino acid biosynthesis and the charging of transfer RNAs, and DapB is involved in the biosynthesis of both lysine and peptidoglycan.)

**Many gaps in amino acid biosynthesis in diverse prokaryotes.** After we updated GapMind based on the genetic data, the amino acid biosynthesis pathways in the 35 bacteria still have 31 gaps: 15 low-confidence steps and 16 medium-confidence steps are on the best paths (Table 1). Six of these gaps are spurious: they reflect errors in the genome sequence or omissions in the protein annotation (Text S3). Another 11 of the 31 gaps are due to diverged enzymes: there is a reasonable candidate for the step that is too diverged from characterized proteins to be called high confidence. In 10 of these cases, the gene appears to be essential (8, 30). In the remaining case, mutants in the gene (N515DRAFT_4305 from *D. japonica*) had low abundance in the pool of transposon mutants, so we were not able to confirm that these mutants were auxotrophic. The remaining 14 gaps indicate novel biology that remains to be discovered (Table 1). We will describe two of these cases in more detail.

First, GapMind did not identify high-confidence candidates for any of the three steps of serine biosynthesis in either *Desulfovibrio vulgaris* Hildenborough or *D. vulgaris* Miyazaki F, which are both strictly anaerobic sulfate-reducing bacteria. In addition, the genetic data did not identify candidate genes for these steps. We also considered whether serine might be formed from glycine: although glycine is usually formed from serine, it might also form by the glycine cleavage reaction in reverse, which may be thermodynamically feasible if one-carbon substrates, such as formate, reach high concentrations (Text S2). However, genes from the glycine cleavage system were not important for the growth of either strain of *D. vulgaris* in minimal media (V. V. Trotter, personal communication, and data from reference 8). A metabolic labeling study also suggests that *D. vulgaris* Hildenborough forms serine from glycolytic intermediates (31), which is consistent with the standard pathway but not with the glycine cleavage reaction in reverse. Thus, serine biosynthesis in *Desulfovibrio vulgaris* remains unresolved.

Second, *B. thetaiotaomicron* does not seem to contain homoserine kinase (ThrB). The curators at TIGRFam proposed that TIGR02535 replaces homoserine kinase by transferring phosphate groups from a donor such as phosphoenolpyruvate to homoserine. TIGR02535 is related to phosphoglycerate mutases, which transfer phosphate groups, and is often adjacent to other genes for threonine synthesis. *B. thetaiotaomicron* does not seem to contain a traditional homoserine kinase, but it does contain a member of TIGR02535 (BT2402). Mutants in BT2402 were important for growth in minimal media unless threonine was added (data from reference 20). However, when BT2402 was introduced into a *thrB* mutant strain of *E. coli*, no growth in minimal medium was observed (Hualan Liu, personal communication). Therefore, it remains uncertain whether BT2402 catalyzes the formation of O-phosphohomoserine or if it has another role in threonine synthesis. Two of the other bacteria we studied genetically, *Phae-*

**TABLE 1** Remaining gaps in amino acid biosynthesis for 35 bacteria that can make all 20 amino acids and have large-scale genetic data

| Type of gap | Pathway: gap | Organism | Comment |
|---|---|---|---|
| Novel | Histidine: HisN | *Synechococcus elongatus* PCC 7942 | Synpcc7942_1763 is a candidate for histidinol phosphatase but is not required for growth |
| Novel | Lysine: DapCE or DapL | *Echinicola vietnamensis* KMM 6221, DSM 17526 | Echvi_3551 is a good candidate for the succinyltransferase DapD, which suggests succinylated intermediates, but DapC and DapE are missing, or Echvi_0124 might be a diverged diaminopimelate aminotransferase (DapL) |
| Novel | Lysine: DapE | *Pedobacter* sp. strain GW460-11-11-14-LB5 | No convincing candidate for the desuccinylase DapE was found |
| Novel | Serine: SerACB | *Desulfovibrio vulgaris* Hildenborough | This genome does not seem to encode the standard SerACB pathway of serine synthesis |
| Novel | Serine: SerACB | *Desulfovibrio vulgaris* Miyazaki F | This genome does not seem to encode the standard SerACB pathway of serine synthesis |
| Novel | Serine: SerB | *Dyella japonica* UNC79MFTsu3.2 | This genome has several weak candidates for phosphoserine phosphatase |
| Novel | Serine: SerB | *Synechococcus elongatus* PCC 7942 | Synpcc7942_2078 is a candidate for phosphoserine phosphatase but is not required growth |
| Novel | Threonine: ThrB | *Bacteroides thetaiotaomicron* VPI-5482 | This genome does not encode a standard threonine synthase or ThrC (BT2401) |
| Novel | Threonine: ThrB | *Dinoroseobacter shibae* DFL-12 | This genome does not encode a standard threonine synthase or ThrC (Dshi_1146) |
| Novel | Threonine: ThrB | *Phaeobacter inhibens* BS107 | This genome does not encode a standard threonine synthase or ThrC (PGA1_c06310) |
| Diverged | Chorismate: AroA | *Echinicola vietnamensis* KMM 6221, DSM 17526 | Echvi_0122 may be a diverged AroA; it appears to be essential |
| Diverged | Cysteine: CysE | *Echinicola vietnamensis* KMM 6221, DSM 17526 | Echvi_0221 may be a diverged serine acetyltransferase; it appears to be essential |
| Diverged | Histidine: HisN | *Desulfovibrio vulgaris* Hildenborough | DVU2940 may be a diverged histidinol phosphatase; it appears to be essential (V. V. Trotter, personal communication) |
| Diverged | Histidine: HisN | *Desulfovibrio vulgaris* Miyazaki F | DvMF_0940 may be a diverged histidinol phosphatase; it appears to be essential |
| Diverged | Histidine: HisC | *Synechococcus elongatus* PCC 7942 | Synpcc7942_1030 may be a diverged histidinol-phosphate aminotransferase; it appears to be essential |
| Diverged | Leu/Ile/Val: IlvI | *Dyella japonica* UNC79MFTsu3.2 | Various strains of *Dyella japonica* have a short IlvI (regulatory subunit of acetolactate synthase), i.e., N515DRAFT_0566 |
| Diverged | Methionine: MetC | *Dyella japonica* UNC79MFTsu3.2 | This organism probably uses the transsulfuration pathway (MetB = N515DRAFT_4363 is important for growth in minimal media); N515DRAFT_4305 is likely to be cystathionine beta-lyase (MetC), but it is also very similar to a cystathionine gamma-lyase (Q5H4T8) |
| Diverged | Phenylalanine: Pdehyd | *Synechococcus elongatus* PCC 7942 | Synpcc7942_0881 may be a diverged prephenate dehydratase; it appears to be essential |
| Diverged | Serine: SerC | *Echinicola vietnamensis* KMM 6221, DSM 17526 | Echvi_1811 may be a phosphoserine aminotransferase; it appears to be essential |
| Spurious | Chorismate: AroL | *Azospirillum brasilense* Sp245 | An open reading frame with 41% identity to AROK_ECOLI is present, but no protein was predicted |
| Spurious | Chorismate: AroC | *Shewanella oneidensis* MR-1 | A frameshift error splits AroC into two reading frames (SO3078.2 and SO_3079) |
| Spurious | Histidine: HisD | *Azospirillum brasilense* Sp245 | A frameshift error in the genome sequence prevented this protein from being predicted (3) |
| Spurious | Histidine: Prs | *Pseudomonas fluorescens* FW300-N1B4 | An open reading frame with 67% identity to KPRS_ECOLI is present, but no protein was predicted |
| Spurious | Methionine: MetZ | *Pseudomonas fluorescens* FW300-N1B4 | The published assembly is missing a region that has an open reading frame with 84% identity to METZ_PSEAE |
| Spurious | Serine: SerC | *Azospirillum brasilense* Sp245 | An open reading frame with 58% identity to SERC_METBF is present, but no protein was predicted |

*obacter inhibens* and *Dinoroseobacter shibae*, also seem to lack homoserine kinase, but they do not contain members of TIGR02535.

Overall, we used the genetic data to reduce the total number of gaps in these 35 bacteria from 130 to 31. Seventeen of the remaining gaps can be explained; the other 14 gaps are due to our limited understanding of amino acid biosynthesis in bacteria.

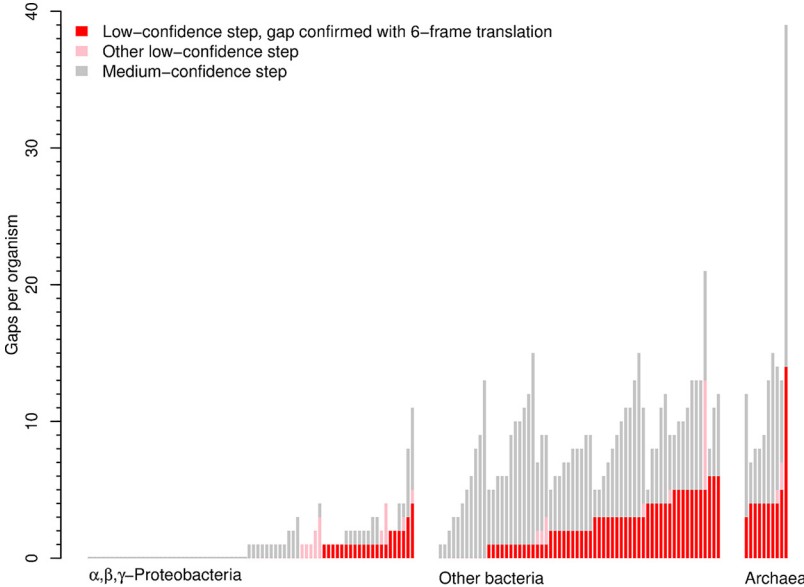

**FIG 5** Number of gaps in amino acid biosynthesis in 148 diverse bacteria and archaea that can grow without amino acids. (These are distinct from the 35 bacteria with fitness data.)

We then tested GapMind on a more diverse collection of 148 bacteria and archaea that can grow in the absence of any amino acids (data from references 32 and 33; Data Set S3). These microbes represent 19 phyla and 132 genera (as classified by GTDB [27]). Most of them are distantly related to the 35 bacteria that we have genetic data for: just 15 of the 148 belong to the same genus, and none of them belong to the same species. Across all pathways, the 148 microbes had an average of 5.2 gaps, including 1.8 low-confidence steps and 3.4 medium-confidence steps (Fig. 5). The most common low-confidence steps were histidinol-phosphate phosphatase (HisN), which was low confidence in 42 organisms (28%), and homoserine kinase (ThrB), which was low confidence in 29 organisms (20%). Of the 29 organisms that seem to lack ThrB, 15 contain the putative alternative enzyme (TIGR02535).

To verify that the low-confidence steps reflect gaps in our knowledge of amino acid biosynthesis, we manually examined a random sample of 20 of them (Data Set S4). None of these steps could be confidently associated with a protein sequence. We did identify potential candidates for 10 of the steps, but the candidates were quite distantly related to characterized proteins (30% identity or less). Just one of these 20 gaps could be explained by a frameshift error or a missing gene call. More broadly, few of the gaps seem to be due to errors in the genome or in protein prediction: when we analyzed the six-frame translation of all 148 genomes, most of the gaps remained. In particular, of 263 gaps that were low-confidence steps when analyzing the annotated proteins, 232 (88%) were still gaps (that is, on the best path but not high confidence) when analyzing the six-frame translation.

We also examined the microbe with the most gaps, which was the hyperthermophilic archaeon *Pyrolobus fumarii* 1A. Although *P. fumarii* can grow with carbon dioxide as the sole source of carbon (34), its amino acid biosynthesis pathways had 14 gaps that were low-confidence steps. Because some steps appear in more than one pathway, these gaps correspond to 11 missing proteins. We manually searched for these 11 missing proteins and found convincing candidates for just three of them (AroD, TrpA, and TrpB). The candidates for AroD (PYRFU_RS04235) and TrpA (PYRFU_RS05090) were already annotated with these functions in RefSeq, but they are too diverged from proteins in GapMind's database to be medium confidence. The candidate for TrpB (PYRFU_RS05405) was annotated as a TrpB-like protein in RefSeq. The other 8 missing proteins are genuine gaps.

Members of the alpha-, beta-, and gammaproteobacteria had far fewer gaps than other bacteria or archaea (Fig. 5). Of 75 alpha-, beta-, and gammaproteobacteria that we analyzed, 37 (49%) had no gaps, while all of the other microbes had at least one gap. We believe that the alpha-, beta-, and gammaproteobacteria have fewer gaps for two reasons: they are the best-studied group of prokaryotes, and they include 29 of the 35 bacteria that have genetic data and that we used to improve GapMind. If we censored GapMind to remove the improvements that we described above, and to ignore biosynthetic proteins that we had previously identified using the genetic data (3), then the number of gaps in these 75 alpha-, beta-, and gammaproteobacteria rose from an average of 1.1 to 3.9. It appears that we have already filled most of the gaps in amino acid biosynthesis in the alpha-, beta-, and gammaproteobacteria, but most other prokaryotes that can grow in minimal media still have unknown steps in their amino acid biosynthesis pathways.

**Identifying known gaps.** In our analysis, we found that the gaps in amino acid biosynthesis pathways were often conserved between related organisms. For 96 of the 148 microbes, the set included another microbe from the same family (as classified by GTDB). We focused on the gaps that were low-confidence steps and that were confirmed by analyzing the six-frame translation. The 96 microbes had 118 such gaps, and 96 of these steps (81%) were also gaps in another microbe from the same family.

The conservation of most gaps implies that known gaps will be useful for understanding other organisms. If a new genome appears to have a gap but is related to an organism that has the same gap and grows in minimal media, then this known gap should not be considered evidence that the organism lacks the pathway. We built a catalog of 257 known gaps in amino acid biosynthesis by combining the 25 genuine gaps in the bacteria with genetic data (Table 1) with the 232 gaps from diverse prokaryotes that were low-confidence steps and were confirmed by analyzing the six-frame translation.

To identify a known gap in a new genome, GapMind compares all of its predicted proteins to the ribosomal proteins from organisms with known gaps. If the median similarity of the ribosomal proteins is above 75% and the related organism has the same gap, then GapMind marks the gap as known. Seventy-five percent similarity across ribosomal proteins corresponds to belonging to roughly the same family in GTDB (see Materials and Methods).

**Tests on bacteria that cannot make all amino acids.** To show that GapMind gives reasonable results for bacteria that cannot synthesize all of the amino acids, we tested it on four bacteria with experimentally determined requirements for one or more of the amino acids: *Lactobacillus helveticus* CNRZ 32, which is auxotrophic for 12 of the 17 amino acids that GapMind represents (35); *Clostridium perfringens* PX7, which is auxotrophic for 11 of the amino acids that GapMind represents (36); *Enterococcus faecalis* V583, which is auxotrophic for seven amino acids (37); and *Clostridium scindens* ATCC 37504, which can synthesize all of the amino acids except tryptophan (38). (*C. perfringens* PX7 was derived from NCTC8798, whose genome is available, by curing a prophage [36]; this is not expected to alter its nutrient requirements.) Across these four bacteria, GapMind classified one or more steps as low confidence for 29 out of the 31 amino acids that are required for growth. In contrast, GapMind identified a low-confidence step(s) for just 2 of 37 amino acids that the four bacteria can synthesize.

The misclassified cases were lysine synthesis in *L. helveticus*, glycine and serine synthesis in *C. perfringens*, and serine synthesis in *E. faecalis*. Although *L. helveticus* requires lysine for growth (35), the biosynthetic pathway appears to be complete except for the acetyl-diaminopimelate aminotransferase DapX; GapMind identified a medium-confidence candidate for DapX. Although *C. perfringens* requires glycine for growth (36), GapMind identified a high-confidence candidate for the serine hydroxymethyltransferase GlyA, which should be sufficient. Conversely, *C. perfringens* grows in the absence of serine, but GapMind identified only low-confidence candidates for SerB (phosphoserine phosphatase) and medium-confidence candidates for SerC (phospho-

## Amino acid biosynthesis in Desulfovibrio alaskensis G20

| Pathway | Steps |
|---------|-------|
| arg | argJ, argB, argC, argD, carA, carB, argI, argG, argH |
| asn | aspS2, gatA, gatB, gatC |
| chorismate | tpiA, fbp, asp_kinase, asd, aroA', aroB', aroD, aroE, aroL, aroA, aroC |
| cys | cysE, cysK |
| gln | gltX, gatA, gatB, gatC |
| gly | glyA |
| his | prs, hisG, hisI, hisE, hisA, hisF, hisH, hisB, hisC, hisN, hisD |
| ile | cimA, leuC, leuD, leuB, ilvI, ilvH, ilvC, ilvD, ilvE |
| leu | ilvH, ilvI, ilvC, ilvD, leuA, leuC, leuD, leuB, ilvE |
| lys | asp_kinase, asd, dapA, dapB, DAPtransferase, dapF, lysA |
| met | asp_kinase, asd, asd_S_transferase, asd_S_ferredoxin, asd_S_perS, metH, B12_reactivation_RamA |
| phe | cmutase, pdehyd, ilvE |
| pro | proB, proA, proC |
| ser | serA?, serC, serB? |
| thr | asp_kinase, asd, hom, thrB, thrC |
| trp | trpE, trpD_1, trpD_2, PRAI, IGPS, trpA, trpB |
| tyr | cmutase, predehyd, tyrB |
| val | ilvH, ilvI, ilvC, ilvD, ilvE |

Confidence: **high confidence** medium confidence low confidence
? – known gap: despite the lack of a good candidate for this step, this organism (or a related organism) performs the pathway

**FIG 6** GapMind's website renders the best paths for amino acid biosynthesis in *Desulfovibrio alaskensis* G20. Each step is color coded by its confidence level, and a question mark indicates known gaps in related organisms.

serine transaminase). Similarly, *E. faecalis* is reported to grow in the absence of serine, but GapMind identified only low-confidence candidates for SerA (3-phosphoglycerate dehydrogenase) or SerB and a medium-confidence candidate for SerC. In a metabolic model of *E. faecalis* (37), only one of the three reactions is present (3-phosphoglycerate dehydrogenase). It is not clear how *C. perfringens* or *E. faecalis* can grow without added serine.

Besides the amino acids that are represented in GapMind, *L. helveticus* also requires glutamate (35), and it requires either aspartate or asparagine unless citrate is provided (39). Citrate can probably alleviate the requirement for aspartate or asparagine because citrate can be cleaved to oxaloacetate (39), which is the carboxylic acid precursor for aspartate. The requirement for glutamate probably indicates that *L. helveticus* cannot make $\alpha$-ketoglutarate (39). Similarly, *C. perfringens* requires aspartate and glutamate (36). Because GapMind does not represent central metabolism, it does not model these dependencies.

**The GapMind website.** At the GapMind website (http://papers.genomics.lbl.gov/ gaps), you can select a genome from various resources, including NCBI's database of assemblies, or you can upload a fasta file of predicted protein sequences. Once you select a genome, the analysis takes about 15 s. Analysis results are stored indefinitely, but if GapMind's database has been updated to include new pathways or enzymes, the analysis will be rerun.

After analysis, the main page for the organism lists the best path for each amino acid (Fig. 6). Gaps are highlighted by color, and known gaps are marked (such as serA or serB in Fig. 6). Each step has hover text with a description of the enzymatic step and the identifier of the top candidate. Clicking on a pathway or step leads to more detailed pages. The page for each step includes how the step was defined and search tools to find additional candidates, including Curated BLAST (40), which can find reading frames that were not annotated. The page for each candidate includes links to tools for analyzing the protein's sequence, including Pfam (15), the conserved domain database (CDD) (41), and PaperBLAST, which finds papers about a protein and its homologs (42). Each candidate's page also includes links to the alignments that led to the identification of the candidate.

## DISCUSSION

GapMind is based on careful curation of a subset of biosynthetic pathways across many prokaryotes that are known to make all 20 amino acids. In contrast to genome-scale metabolic modeling, a pathway-centric approach allows curation effort to focus on the reactions that are most relevant to the capabilities of interest. Because of this, GapMind implicitly assumes that all intermediates in central metabolism are available. This is likely to be true if the microbe contains most of the amino acid biosynthesis pathways, but it might not be true for microbes that have many auxotrophies, such as *Lactobacillus helveticus* CNRZ 32. GapMind also assumes that other amino acids are available, but if this is not likely to be the case, it should be obvious from GapMind's results.

GapMind relies on the predicted proteins in the genome annotation. Omissions in the list of predicted proteins or errors in the genome sequence sometimes lead to spurious gaps. For most of the individual steps, the GapMind website provides links to Curated BLAST for Genomes, which can find candidates that have frameshifts or were not annotated as proteins (40). Curated BLAST can also be useful for finding highly diverged candidates that are less than 30% identical to the characterized or curated proteins for that step (especially if there is no TIGRFam for that step).

GapMind uses the similarity of protein sequences to rate the confidence of candidates. Many other features could be used. In particular, we did not incorporate specificity-determining residues (e.g., see references 28 and 43) into GapMind, because this information is available for few enzyme families. GapMind also does not consider whether a candidate gene clusters with other proteins in the pathway. Genomic context may be taken into account indirectly via the curation effort behind TIGRFam or Swiss-Prot (although Swiss-Prot annotations never lead to high-confidence assignments in GapMind unless they are based on experimental evidence).

The GapMind code should be suitable for reconstructing other metabolic capabilities, such as vitamin synthesis or sugar catabolism. Adding new pathways requires curation effort to describe multisubunit enzymes and to describe reactions that do not have EC numbers. When adding a new pathway, it is also important to check the quality of the results and to identify enzymes with ambiguous or incorrect descriptions that should be ignored. This could be partially automated if the growth capabilities and the genomes of many microbes were available.

In conclusion, GapMind quickly identifies potential pathways for amino acid biosynthesis in a microbial genome. For most bacteria that can synthesize all 20 amino acids, GapMind identifies just a few missing steps or gaps, and the GapMind website provides interactive tools to investigate these gaps. To indicate that a gap may correspond to novel biology (instead of a missing capability), GapMind reports if a related microbe that has the same gap is known to grow in minimal media. To fill some of the gaps in our understanding of amino acid biosynthesis, we tested a preliminary version of GapMind against diverse bacteria with genetic data. We identified additional genes involved in arginine synthesis with succinylated intermediates in *Bacteroidetes*, we proposed that *Dyella japonica* synthesizes tyrosine from phenylalanine, and we annotated dozens of divergent enzymes based on genetic data. However, we still do not understand how most bacteria or archaea can make all 20 amino acids.

## MATERIALS AND METHODS

**Data sources.** Pathways were taken from MetaCyc's website (accessed January to April 2019).

Experimentally characterized proteins were taken from the curated part of PaperBLAST's database in January 2019. PaperBLAST only incorporates the subset of Swiss-Prot with experimental evidence, but some of these proteins only have evidence as to their expression, not their function. EcoCyc (44) and CharProtDB (45) also contain significant numbers of uncharacterized proteins. Proteins were deemed uncharacterized and were filtered out if the description began with "uncharacterized" or matched "uncharacterized protein," "DUFnnnn family protein," "PFnnnnn family protein," or "UPFnnnnn family protein." For EcoCyc and CharProtDB, descriptions beginning with "putative" or "protein" also were filtered out, and for CharProtDB, descriptions beginning with "probably" were filtered out. This left 113,710 different sequences: 84,815 from Swiss-Prot, 21,497 from BRENDA (46), 8,629 from CAZy (47), 7,397 from CharProtDB, 6,474 from MetaCyc, 3,441 from EcoCyc, 2,749 from REBASE (48), and 1,319

reannotations based on genetic data from the Fitness Browser (8). (These numbers sum to more than the number of sequences because of overlap between databases.) Previously filled gaps in amino acid biosynthesis (3) were incorporated via the Fitness Browser's reannotations.

Curated proteins were taken from Swiss-Prot (downloaded in April 2019). To keep the database small, we only used proteins from bacteria or archaea that were annotated as enzymes (with an EC number, even if incomplete). We excluded fragment proteins and proteins with "CAUTION" comments (which indicate uncertainty as to whether the annotation is correct). We then clustered the sequences at 60% identity (using usearch -cluster_fast). If the EC assignments within a cluster varied, we split the cluster by EC number. We then arbitrarily selected one sequence from each cluster, giving a secondary database of 45,090 curated sequences.

As sources of protein families, we used the most recent release of TIGRFam (15.0) and Pfam release 32.0 (from September 2018).

Fitness data were taken from the Fitness Browser (http://fit.genomics.lbl.gov).

**Microbes that can make all of the amino acids.** For the 35 bacteria with genetic data, we and our colleagues have grown 34 of them in minimal media with no amino acids present. For *Bacteroides thetaiotaomicron* VPI-5482, our defined medium includes cysteine and methionine, but these are not required for growth by this strain (49).

To identify genome sequences for additional microbes that can make all of the amino acids, we used two sources: a comparative genomics analysis of nitrogen-fixing bacteria and archaea (32) and the KOMODO database of organism-medium pairings (33). Using the strain-level identifiers, we were able to link 63 nitrogen-fixing genomes (32) to assemblies in RefSeq. We manually removed the endosymbiont UCYN-A; the other organisms are all believed to grow in minimal media.

From KOMODO, we identified a subset of media that did not contain amino acids or undefined components, such as yeast extract or Casamino Acids. Although KOMODO reports an "IsComplex" field for media, this field is not sufficient, because media could contain individual amino acids. The DSMZ's instructions for growing some organisms also state that yeast extract should be added even if the base medium is defined. To filter out these cases, we searched through the PDF instructions associated with each medium. We also removed from consideration any organisms whose genomes were not in RefSeq or whose genomes had over 200 scaffolds, as well as a few genomes that had been sequenced with Ion Torrent and appeared to have many frameshift errors. This left 88 genomes for microbes that grow in minimal media.

After removing a few overlaps between the nitrogen-fixing or KOMODO organisms or with the bacteria that have genetic data, we were left with genomes for 148 bacteria and archaea that can make all of the amino acids (see Data Set S3 in the supplemental material). To classify these microbes into phyla, families, and genera, we used GTDB release 89.0 (https://data.ace.uq.edu.au/public/gtdb/data/releases/release89/89.0/).

We believe that these 148 genomes are of high quality and that few of the gaps are due to errors or omissions in the genome sequences. Ninety-two of the 148 genomes are classified as complete in NCBI's assembly database, and complete genomes had about the same number of low-confidence steps, on average, as incomplete genomes (1.9 versus 1.6, respectively; $P = 0.39$, $t$ test). The organism with the largest number of gaps (*P. fumarii* 1A) also has a complete genome. Furthermore, as discussed in Results, few of the gaps seem to be due to frameshift errors, and most of the gaps were conserved in another bacterium (if there was a relative among these 148 microbes).

**Where should each pathway begin?** Most of GapMind's pathways begin with central metabolic intermediates or with other amino acids. The central metabolites include the 13 central metabolites, as defined in MetaCyc, as well as isocitrate (as a precursor to glyoxylate and glycine). There are a few other precursors whose biosynthesis is complex and is not represented in GapMind: ATP (a precursor to histidine), methyltetrahydrofolate or methyl corrinoid proteins (which are precursors to methionine), and propionate (a precursor to isoleucine). Finally, homocysteine and phosphoserine are intermediates in the biosynthesis of methionine and serine (respectively) but also can be precursors to cysteine (see "Dependencies between pathways," below). GapMind does not represent central metabolism or the regeneration of cofactors such as ATP or NAD(P)H.

**Defining each step.** Each step is defined by one or more EC numbers, terms, or UniProt identifiers. EC numbers can be matched to curated descriptions and to families in TIGRFam. EC numbers work well for most steps, but some steps do not have fully specified four-digit EC numbers or are catalyzed by heteromeric protein complexes. Thus, steps can also be defined by terms that appear in the curated protein descriptions. For instance, imidazole glycerol phosphate synthase (an enzyme in histidine biosynthesis) is a heterodimer and described as two steps, hisF and hisH. hisF is defined by the curated term "hisF" or by TIGRFam TIGR00735. GapMind's matching of terms to curated descriptions is case-insensitive, and each match must begin and end at word boundaries. For some steps, we also identified specific sequences (by UniProt identifier) that are known to perform the step but are not curated in the databases that GapMind relies on. We identified 99 such sequences, mostly by using the fitness data but also from the literature.

Because enzyme subunits may not be described consistently across the databases that GapMind relies on, the definition of a step can also "ignore" proteins that might or might not match the step. Hits to ignored proteins are disregarded when testing if a candidate is similar to proteins with other functions. Ignore can also be useful when closely related proteins have different substrate specificities. For example, 3-isopropylmalate dehydratase (LeuCD) from *Desulfovibrio vulgaris* Hildenborough (DVU2982 and DVU2983) is over 50% identical to a 2,3-methylmalate dehydratase (UniProt accession numbers Q0QLE2 and Q0QLE1), which would lead to both LeuC and LeuD being moderate-confidence candidates. Fitness

data confirm that DVU2982 and DVU2983 are required for amino acid biosynthesis (V. V. Trotter, personal communication). Thus, we modified the step definitions for LeuC and LeuD to ignore Q0QLE2 and Q0QLE1. As another example, it can be difficult to distinguish O-acetylhomoserine sulfhydrylase and O-succinylhomoserine sulfhydrylase. (In fact, misannotation of these enzymes is widespread [43].) Hits to proteins annotated with EC 2.5.1.49 (O-acetylhomoserine sulfhydrylase) are ignored when determining if a candidate for O-succinylhomoserine sulfhydrylase should be considered high confidence. This example also illustrates that although GapMind tries to find a high-confidence path, it may not be confident as to the cofactors or even the exact substrates. Similarly, in Results, we mentioned the difficulty of distinguishing prephenate dehydrogenase and arogenate dehydrogenase.

GapMind's pathways include 183 total steps, but some of these steps are identical (or nearly so) across different pathways. Not considering these identical steps, there are 149 different steps represented in GapMind. After removing heteromers, multicomponent enzymes, or carrier proteins, this reduces to 133 enzymes. If different versions of an enzyme have different subunit compositions, they are described separately, so these correspond to 130 different reactions.

**Finding candidates for each step.** Each step definition is converted to a list of characterized proteins, uncharacterized but curated proteins, and/or protein families. GapMind then uses ublast (16) to compare the predicted proteins in the genome to the characterized or curated proteins. It considers hits with at least 30% identity and with an E value of <0.01. GapMind uses HMMER 3 (17) to compare the predicted proteins to families and uses the trusted cutoff provided by the curator of each family.

**Scoring candidates for each step.** GapMind then checks if these candidates are similar to proteins that have other functions. Specifically, it compares each candidate in the genome of interest (whether from ublast or HMMER) to the database of characterized proteins, again using ublast with at least 30% identity and an E value of <0.01. Any similarity between a candidate and a protein that matches the step or is ignored for that step is disregarded. To support the identification of fusion proteins, hits that do not overlap at least 50% of the relevant region of the candidate (that was identified by ublast or HMMER) also are ignored. The remaining hit with the highest bit score (if any) is the "other" hit.

A candidate for a step is considered high confidence if it is over 40% identical to a characterized protein, the alignment covers over 80% of that protein, and the bit score is at least 10 bits higher than that for the other hit. Alternatively, a candidate is high confidence if HMMER finds a hit (above the trusted cutoff), the alignment covers at least 80% of the HMM, and the other hit is under 40% identity or has under 75% coverage. A candidate for a step is medium confidence if it is over 40% identical to either a characterized or curated protein with above 70% coverage (regardless of the other hit), or is above 30% identical to a characterized or curated protein with above 80% coverage, and the bit score is higher than that for the other hit or if HMMER finds a hit (above the trusted cutoff). Other hits from ublast with at least 50% coverage are low confidence.

**Split candidates.** GapMind attempts to join low-coverage hits from ublast together if the alignments score noticeably higher than other hits (by at least 10 bits) and they are similar to the same characterized or curated protein. GapMind checks that there is little overlap between the alignments (at most 20% of either alignment) and that the combined alignment covers at least 70% of the characterized or curated protein. If the split candidate (the combination of the two alignments) has a higher confidence score (as defined above) than either of the components, then the split is chosen as the candidate instead.

On the GapMind website, split candidates are marked with an asterisk. If the two parts of the split are adjacent, it is often ambiguous whether the protein-coding gene is actually split, disrupted by a genuine frameshift, or disrupted by a frameshift error in the genome sequence.

For the 35 organisms with genetic data, we identified 11 cases where the only high-confidence candidate for a step was a split protein. All of these were on the best path for that amino acid. Nine of these cases involved MetH, and we believe that these are genuine splits because similar splits are found in related genomes. The other two cases may be spurious. In *Pseudomonas fluorescens* FW300-N1B4, phosphoribosylanthranilate isomerase (a step in tryptophan synthesis) appeared to be split into two adjacent proteins in the public assembly (GCF_001625455.1), which is based on PacBio and Illumina data. However, in an alternative assembly based on the Illumina data only, there is a single-nucleotide insertion in this region (5 Cs instead of 4 Cs starting at position 4993 of GenBank accession no. NZ_LUKJ01000003.1). This change leads to a single reading frame, so the split is probably spurious. Finally, in *Paraburkholderia bryophila* 376MFSha3.1, threonine ammonia-lyase (IlvA) was identified as split into two proteins (H281DRAFT_04606 and H281DRAFT_01887). The first protein contains a pyridoxal-phosphate-dependent enzyme domain (PF00291), and the second protein contains two copies of the C-terminal regulatory domain of threonine dehydratase (PF00585). Neither protein has strong auxotrophic phenotypes, which indicates genetic redundancy with H281DRAFT_04028, predicted to be a catabolic threonine dehydratase. One part of the split (H281DRAFT_04606) is over 70% identical to the N-terminal half of HSERO_RS19510 from *Herbaspirillum seropedicae* SmR1, which does have auxotrophic phenotypes. It is not clear if *P. bryophila* has a split IlvA or the catalytic domain alone (H281DRAFT_04606) is sufficient for activity.

**Scoring pathways.** The score for a step is the score of its best candidate (high, medium, or low). The score for a pathway is the lowest score of any of its steps (or subpathways). The best path for an amino acid is the one that gives the best score. If two paths have the same score, then GapMind considers a secondary score that gives weights of −2, −0.1, and +1 to low-, medium-, and high-confidence steps. If there is still a tie, then GapMind chooses the longer path.

**Dependencies between pathways.** To indicate dependencies between pathways, GapMind includes requirements that link a pathway or subpathway to a step in the synthesis of another amino acid that must be present (or must not be present). If these requirements are violated, then GapMind issues

a warning. We chose not to give amino acids as requirements (such as a serine requirement for cysteine biosynthesis), because GapMind already shows if an amino acid might be required for growth. However, we do use requirements to define dependencies on intermediates. For example, some organisms form cysteine from phosphoserine instead of from serine; if this pathway is on the best path, then GapMind will check if serA and serC are present. As another example, GapMind will issue a warning if the organism is predicted to synthesize methionine from cysteine (transsulfuration) and also cysteine from methionine (reverse transsulfuration). This hypothetical organism might require either methionine or cysteine for growth because it might not be able to assimilate sulfide.

**Similarity to microbes with known gaps.** To quickly identify similarities between a genome of interest and the microbes that have known gaps, GapMind relies on ribosomal proteins as marker genes. Specifically, it uses the ribosomal subset of the marker genes used in GTDB (25 for bacteria and 32 for archaea). There are 93 microbes with known gaps (9 from the 35 bacteria with genetic data and 84 from the 148 diverse prokaryotes that grow in minimal media). For each of the 93 microbes with a known gap, we used HMMER (with the models specified by GTDB) to identify the ribosomal proteins. If a genome contains more than one protein matching an HMM, all are ignored.

When analyzing a new genome, GapMind uses usearch with global alignment (16) to quickly find proteins in the new genome that are at least 50% identical to the marker genes and with alignment coverage of at least 70%. GapMind only searches for the top 20 hits (-maxaccepts 20 -maxrejects 20). Only one-to-one hits are retained. Genomes are considered to be related if there at least 10 retained hits and the median hit is at least 75% identical. We tested this definition of "related" by comparing the marker genes from the 93 microbes that have known gaps to each other. Of the 94 pairs of microbes from the same family (as classified in GTDB), 72 were related. Of the 4,184 pairs of microbes that do not belong to the same family, just 31 were related.

**Software.** GapMind is written in Perl 5. The Web-based interface relies on the common gateway interface library (CGI.pm). We used usearch/ublast 10.0 (the free 32-bit version) and HMMER 3.1b2. The Web server runs usearch and HMMER with 6 threads.

**Data availability.** The code for GapMind is included in the PaperBLAST code repository (https://github.com/morgannprice/PaperBLAST). The definition of each pathway, with comments, is included in the code repository in the gaps/aa subdirectory. That subdirectory also includes tables of dependencies between pathways, curated gaps (in the 35 bacteria), and known gaps (in the 148 diverse microbes). The database of characterized proteins and the list of proteins associated with each step are available for download (http://papers.genomics.lbl.gov/tmp/path.aa/aa.resources.tar.gz). The code, the database, and the results (for the 35 bacteria and the 148 diverse microbes) are also archived at figshare (https://doi.org/10.6084/m9.figshare.9693689.v1). (The figshare also includes results for two additional genomes that were removed from our final analysis because they have over 200 scaffolds: *Nocardiopsis lucentensis* and *Thauera aminoaromatica*.) The fitness data are available from the Fitness Browser (http://fit.genomics.lbl.gov).

## SUPPLEMENTAL MATERIAL

Supplemental material is available online only.
**TEXT S1**, PDF file, 0.1 MB.
**TEXT S2**, PDF file, 0.04 MB.
**TEXT S3**, PDF file, 0.04 MB.
**DATA SET S1**, XLS file, 0.01 MB.
**DATA SET S2**, XLS file, 0.03 MB.
**DATA SET S3**, XLS file, 0.04 MB.
**DATA SET S4**, XLS file, 0.02 MB.

## ACKNOWLEDGMENTS

We thank Valentine V. Trotter for prepublication access to genetic data for *Desulfovibrio vulgaris* Hildenborough.

This work was performed by ENIGMA, a Scientific Focus Area Program supported by the U.S. Department of Energy, Office of Science, Office of Biological and Environmental Research, Genomics:GTLFoundational Science through contract DE-AC02-05CH11231 between Lawrence Berkeley National Laboratory and the U.S. Department of Energy.

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
