## [Reviewer comments · mSystems]

GapMind: Automated annotation of amino acid biosynthesis

Morgan Price, Adam Deutschbauer, and Adam Arkin

Corresponding Author(s): Morgan Price, Lawrence Berkeley National Laboratory

Review Timeline:

Submission Date:	March 30, 2020
Editorial Decision:	May 15, 2020
Revision Received:	June 4, 2020
Accepted:	June 5, 2020

Editor: Steven Hallam

Reviewer(s): The reviewers have opted to remain anonymous.

Transaction Report:

DOI: <https://doi.org/10.1128/mSystems.00291-20>

May 15, 2020

Mr. Morgan N Price
Lawrence Berkeley National Laboratory
Environmental Genomics and Systems Biology Division
1 Cyclotron Road
Berkeley

Re: mSystems00291-20 (GapMind: Automated annotation of amino acid biosynthesis)

Dear Mr. Morgan N Price:

Thank you for submitting your paper to mSystems. The reviewers found the work interesting, relevant and usable. From an editorial perspective it was a pleasure to handle the manuscript which is clearly written and addresses an important area of research. Reviewer #1 pointed out some minor modifications that should be relatively easy to make. Interestingly both reviewers (in different) ways made a case for adding use cases. Reviewer #1 suggested a human microbiome example which would be on the more complex end of the spectrum while Reviewer #2 indicated application to endosymbionts with known gaps in their amino acid biosynthetic pathways. Providing such use cases would likely broaden the appeal of the web server to more researchers.

We look forward to seeing a revised version of the manuscript soon.

Below you will find the comments of the reviewers.

To submit your modified manuscript, log onto the eJP submission site at <https://msystems.msubmit.net/cgi-bin/main.plex>. If you cannot remember your password, click the "Can't remember your password?" link and follow the instructions on the screen. Go to Author Tasks and click the appropriate manuscript title to begin the resubmission process. The information that you entered when you first submitted the paper will be displayed. Please update the information as necessary. Provide (1) point-by-point responses to the issues raised by the reviewers as file type "Response to Reviewers," not in your cover letter, and (2) a PDF file that indicates the changes from the original submission (by highlighting or underlining the changes) as file type "Marked Up Manuscript - For Review Only."

Due to the SARS-CoV-2 pandemic, our typical 60 day deadline for revisions will not be applied. I hope that you will be able to submit a revised manuscript soon, but want to reassure you that the journal will be flexible in terms of timing, particularly if experimental revisions are needed. When you are ready to resubmit, please know that our staff and Editors are working remotely and handling submissions without delay. If you do not wish to modify the manuscript and prefer to submit it to another journal, please notify me of your decision immediately so that the manuscript may be formally withdrawn from consideration by mSystems.

To avoid unnecessary delay in publication should your modified manuscript be accepted, it is important that all elements you upload meet the technical requirements for production. I strongly recommend that you check your digital images using the Rapid Inspector tool at <http://rapidinspector.cadmus.com/RapidInspector/zmw/>.

Sincerely,

Steven Hallam

Editor, mSystems

Journals Department
Reviewer comments:

Reviewer #1 (Comments for the Author):

The manuscript by Price and colleagues reports a web tool for annotation of amino acid biosynthetic pathways in bacteria. This is a highly relevant problem, especially in the context of biotechnology (e.g. to optimize growth media or to identify gene KO targets) and microbial communities (amino acids are among the most prevalent group of compounds that are exchanged between species). We have a few suggestions/comments towards improving the clarity and making the tool more widely used.

1. As a majority of the use is likely to be for bacteria that are auxotrophic, it will be good to include some benchmarking cases beyond the 35 chosen (which can grow without AA supplementation). Gut bacteria could be a highly useful case (for defined media for some gut bacteria, see: Tramontano et al., Nat Microbiol 2018).
2. The Abstract ends on a highly technical note - a summary/forward-looking sentence would fit better.
3. The authors mention that other tools that predict AA auxotrophies (like the IMG database) have multiple limitations, and then go through the curation of the pathways one by one, which is relevant, but a results overview figure with a systematic benchmark against IMG would be appropriate.

Reviewer #2 (Comments for the Author):

GapMind will be a very useful tool for anyone working on bacterial genome annotation, or for people interested in amino acid metabolism. Also, the paper was fun and interesting to read.

I work on very weird, very divergent bacteria that are endosymbionts of animals. I ran GapMind on some of my strangest genomes, genomes on which I had previously performed a very careful annotation of the amino acid pathways by hand, and GapMind performed perfectly. I think endosymbionts are pretty tough test cases, and the software was very fast and very accurate. What more could you ask for?

Comments from Reviewer #1

1. As a majority of the use is likely to be for bacteria that are auxotrophic, it will be good to include some benchmarking cases beyond the 35 chosen (which can grow without AA supplementation). Gut bacteria could be a highly useful case (for defined media for some gut bacteria, see: Tramontano et al., Nat Microbiol 2018).

The section "Tests on bacteria that cannot make all amino acids" already included two gut bacteria whose amino acid requirements are known (*Enterococcus faecalis* and *Clostridium scindens*). From reading Tramontano et al and its references, we identified another one, *Clostridium perfringens*, which we added to our analysis. Tramontano et al reported that dozens of other gut bacteria grow in defined media, but they did not identify the exact amino acid requirements for any of those strains. For instance, if they found that a bacterium grows in a defined medium with six amino acids, we don't know which of those six amino acids are required for growth; it's possible that just one is required. It is more straightforward and informative to assess GapMind's results when the exact requirements for growth are known.

Tramontano et al also reported that automatically generated metabolic models usually fail to predict growth in defined media (that the bacteria do grow in). This is mentioned in the revised introduction.

2. The Abstract ends on a highly technical note - a summary/forward-looking sentence would fit better.

We revised the end of the abstract to focus on why we think identifying "known" gaps is important, and we added a sentence about the potential impact of GapMind.

3. The authors mention that other tools that predict AA auxotrophies (like the IMG database) have multiple limitations, and then go through the curation of the pathways one by one, which is relevant, but a results overview figure with a systematic benchmark against IMG would be appropriate.

We agree that it might be better to test IMG's tool on the same organisms as in the present study. However, it seems that the IMG phenotype predictions are no longer being maintained. They are no longer included on IMG's genome overview page, and we did not find a way to navigate from the genome overview page to the predictions. There is a page that lets you view their pre-computed predictions across all prokaryotes, but these are apparently out of date: "This option uses pre-computed results. It is fast, but does not reflect recent database changes." And the page to view those thousands of predictions as a table crashes or times out. Overall -- it might be possible to analyze the IMG predictions on a large scale, but since they are no longer maintained, it didn't seem appropriate.

As mentioned in the introduction, a previous publication of ours did include a test of IMG's tool on 10 diverse bacteria that grow in minimal media (ref. 3). These 10 bacteria are a subset of the 35 bacteria with fitness data that were used to test GapMind. The IMG predictions had an incredibly high rate of false predictions of auxotrophies, with an average of 6 per genome.

June 5, 2020

Mr. Morgan N Price
Lawrence Berkeley National Laboratory
Environmental Genomics and Systems Biology Division
1 Cyclotron Road
Berkeley

Re: mSystems00291-20R1 (*GapMind*: Automated annotation of amino acid biosynthesis)

Dear Mr. Morgan N Price:

Your manuscript has been accepted, and I am forwarding it to the ASM Journals Department for publication. For your reference, ASM Journals' address is given below. Before it can be scheduled for publication, your manuscript will be checked by the mSystems senior production editor, Ellie Ghatineh, to make sure that all elements meet the technical requirements for publication. She will contact you if anything needs to be revised before copyediting and production can begin. Otherwise, you will be notified when your proofs are ready to be viewed.

Sincerely,

Steven Hallam
Editor, mSystems

Journals Department
S4: Accept
S2: Accept
S3: Accept
S1: Accept
S1: Accept
S2: Accept
S3: Accept